# Caspase 3 and Cleaved Caspase 3 Expression in Tumorogenesis and Its Correlations with Prognosis in Head and Neck Cancer: A Systematic Review and Meta-Analysis

**DOI:** 10.3390/ijms231911937

**Published:** 2022-10-08

**Authors:** Fábio França Vieira e Silva, María Elena Padín-Iruegas, Vito Carlo Alberto Caponio, Alejandro I. Lorenzo-Pouso, Paula Saavedra-Nieves, Cintia Micaela Chamorro-Petronacci, José Suaréz-Peñaranda, Mario Pérez-Sayáns

**Affiliations:** 1Oral Medicine, Oral Surgery and Implantology Unit (MedOralRes), Faculty of Medicine and Dentistry, University of Santiago de Compostela, 15782 Santiago de Compostela, Spain; 2ORALRES Group, Health Research Institute of Santiago de Compostela (FIDIS), 15782 Santiago de Compostela, Spain; 3Human Anatomy and Embriology Area, Departament of Funcional Biology and Health Sciences, University of Vigo, 36310 Vigo, Spain; 4Department of Clinical and Experimental Medicine, University of Foggia, 71100 Foggia, Italy; 5Department of Statistics and Operative Investigation, University of Santiago de Compostela, 15782 Santiago de Compostela, Spain

**Keywords:** head and neck cancer, oral premalignant disorders, Caspase 3, cleaved Caspase 3, prognosis

## Abstract

Head and neck cancer (HNC) is an ascending and agressive disease. The search for new molecular markers is emerging to solve difficulties in diagnosis, risk management, prognosis and effectiveness of treatments. Proteins related to apoptotic machinery have been identified as potential biomarkers. Caspase 3 is the main effector caspase and has a key role in apoptosis. The objective of this systematic review and meta-analysis is to review studies that analyze changes in Caspase 3 and Cleaved Caspase 3 expression both in oral premalignant disorders (OPMD) as well as in head and neck cancer (HNC). This study also proposes to review the prognostic values associated with HNC according to the expression of Caspase 3. Medline (via PubMed), EMBASE, Scopus, Cochrane, Web of Science and Grey Literature Database were screened from inception to june of 2022 and 18 studies were selected and 8 were included in the prognostic meta-analysis. Results related to the comparison of Caspase 3 expression demonstrated similar expression of Caspase 3 in HNC, with an average of 51.9% (9.5–98.1) showing high/moderate expression compared to 45.7% (14.6–84.7) in OPMD. Of interest, Cleaved Caspase 3 resulted incresed in HNC when compared with OPMD, being 73.3% (38.6–88.3) versus 22.9% (7.1–38.7). Pooled Fixed effect of HR values (95% CI) for OS related to Caspase 3 IHC expression in HNC patients was 1.48 (95% CI 0.95–2.28); also, the rate of heterogeneity was low, as revealed by I^2^ = 31%. For DFS was 1.07 (95% CI 0.79–1.45) with I^2^ = 0% and DSS showed a HR of 0.88 (95% CI 0.69–1.12) with I^2^ = 37%. Caspase 3 and Cleaved Caspase 3 expression could be linked with malignancy progression, but the expression of Caspase 3 did not influence the prognosis of patients with HNC.

## 1. Introduction

Head and neck cancer (HNC) is an increasingly common disease and includes tumors usually arising from different epithelia, such as the upper aerodigestive tract, the lips, oral cavity (mouth), nasal cavity, pharynx, larynx and paranasal sinuses [1,2,3,4]. These neoplasms exhibit an aggressive malignancy profile characterized mainly by the wide potential for invasion of adjacent tissues and metastases to distant organs, even in early stages [5,6].

The search for new molecular markers has emerged to solve the difficulties in diagnosing, determining the degree of risk, predicting the prognosis and evaluating the effectiveness of treatment [7,8,9]. Increased knowledge of the molecular mechanisms involved in cancer development has allowed the development of molecular treatment strategies to restore normal tumor suppressor gene function in the tumor or disrupt intracellular pathways that transcribe aberrant growth signals, among other goals [10].

Among commonly reported aberrant pathways, apoptosis has emerged as a hallmark in cancer biology [11,12,13]. Apoptosis is a genetic cell death program that functions primarily to eliminate senescent or altered cells that are useless or harmful in a multicellular organism [11,14,15,16,17,18,19]. An imbalance of apoptotic pathways is one of the fundamental processes in carcinogenesis [17,20,21,22,23,24,25]. Irreparable DNA damage must be accurately recognized in order to eliminate the damaged cells, but cancer cells are able to evade growth inhibition signals and develop mechanisms to avoid apoptosis even in the face of this damage [10,11,13,26].

The apoptotic machinery has main upstream components, the so-called regulatory agents, and downstream ones, so-called effector agents. These regulators are part of two main circuits: one responsible for receiving and processing extracellular death-inducing signals (the extrinsic apoptotic program) and the other responsible for detecting and integrating intracellular signals (the intrinsic program) [22,25,27,28,29,30,31,32]. Both end up generating the activation of a cascade of proteases called caspases (cysteine-containing aspartic acid-specific proteases) that are synthesized as proenzymes, activated by proteolytic cleavage, which may cleave other caspases as part of the apoptotic signaling cascade [21,22,25,27,29,30,33,34,35,36,37,38,39,40]. Initiator caspases (including Caspase 2, 8, 9 and 10), upon activation by pro-apoptotic signals, cleave and activate effector caspases (including Caspase 3, 6 and 7) [14,41,42,43,44,45,46,47,48,49].

Caspase 3 is responsible for nuclear alterations in apoptosis, and it is considered the main effector caspase [50,51,52,53] since both extrinsic and intrinsic pathways lead to Caspase 3 activation [27,54,55,56] (Figure 1).

Caspase 3 is cleaved at an aspartate residue to produce a p12 and p17 subunit to form the cleaved Caspase 3, which is responsible for morphological and biochemical changes in apoptosis [57,58,59,60] (Figure 1). Caspase 3 is the most downstream enzyme in the apoptosis-inducing protease pathway, and is probably the most clearly associated with cell death, as it cleaves key proteins in the cell repair process [37,61,62,63,64,65,66,67].

The major aim of our systematic review and meta-analysis is to review studies that describe changes through the immunohistochemistry (IHC) of Caspase 3 and cleaved Caspase 3 expression both in oral potentially malignant disorders (OPMD) and HNC, investigating whether there is a significant difference in the expression of this protein between the two groups. In addition, we also propose to review the prognostic values associated with the expression of Caspase 3 in patients with HNC. 

This systematic review and meta-analysis is of paramount importance, as it is the first to analyze the expression of Caspase 3 and cleaved Caspase 3 in HNC and OPMD, as well as the association of Caspase 3 expression with the prognosis of these patients. The results are important for considering the possibility of a failure in the cleavage of Caspase 3 and the impossibility of its activation in the form of cleaved Caspase 3, which could favor the tumorigenic process. In addition, we will consider the implications of a low expression in this protein, which, because it is an effector caspase, would be directly associated with a possible failure in the apoptotic machinery, allowing the continuous proliferation of cancer cells and favoring the development of the disease. Another aspect of critical importance to be explored is whether there is any relationship between the expression of this protein and the patient’s prognosis.

## 2. Material and Methods

The protocol of this systematic review was previously designed by FFV and agreed upon by all authors and registered in PROSPERO (CRD42022341672). The present systematic review and meta-analysis was performed according to Preferred Reporting Items for Systematic Reviews and Meta-Analyses (PRISMA) and the guidelines of the Meta-analysis of Observational Studies in Epidemiology (MOOSE) group. 

The search question was formulated according to the PICO framework as follows:

“Is Caspase 3 protein expression altered in sample of patients with HNC and OPMD? If so, did this change affect the patient’s prognosis?” According to the PICO method: population (patients with HNC or OPMD), intervention (IHC quantifying Caspase 3 expression), comparison (high or low categorization for Caspase 3 expression) and outcome (long-term outcomes, prognosis, such as overall survival, disease-free survival or disease-specific survival). The same PICO framework was applied to investigate cleaved Caspase 3.

### 2.1. Search Strategy 

Medline (via PubMed), EMBASE, Scopus, Cochrane and Web of Science were searched from inception to June 2022. The last search was performed on 25 July for the screening of new published investigations. The Grey Literature Database (accessed on July 2022) was screened at the New York Academy of Medicine. Searches were conducted by combining thesaurus terms used by the databases (accessed on July 2022) (e.g., MeSH and EMTREE) and free text words. For Medline, the following algorithm was used: (“Head Neck Squamous Cell Carcinoma” [MeSH Terms] OR (“Head Neck Cancer” [All Fields] OR (“Oral Squamous Cell Carcinoma” [All Fields] OR (“Oral Cancer” [All Fields] OR (“Premalignant Lesions” [All Fields] OR (“Oral Lichen Planus” [All Fields] OR (“Oral Leukoplakia” [All Fields] AND “CASP3” [All Fields]) OR “Caspase-3” [All Fields] OR “Caspase 3” [All Fields] AND (“Prognosis” [MeSH Terms] OR “Prognostic” [All Fields] OR “Survival” [All Fields]). The aforementioned syntax was conveniently adapted for each database (accessed on July 2022).

This search strategy was coupled with manual searching in the following journals related to oral medicine, oral/maxillofacial surgery, oral pathology and oncology: *Anticancer Research*; *Archives of Oral Biology*; *Clinical Oral Investigations*; *European Archives of Otorhinolaryngology*; *European Journal of Oral Sciences*; *Head & Neck*; *International Journal of Oral and Maxillofacial Surgery*; *International Journal of Oral Science*; *Journal of Craniomaxillofacial Surgery*; *Journal of Oral and Maxillofacial Surgery*; *Journal of Oral Pathology and Medicine*; *Oral Diseases*; *Oral Oncology*; *Oral Surgery*; *Oral Medicine*; *Oral Pathology*; and *Oral Radiology*. Potentially relevant articles that any of the authors were familiar with, as well as reference lists from the retrieved articles, were also comprehensively checked. 

All references retrieved were managed using the software Mendeley Desktop v1.19.8 (Elsevier, London, UK), and duplicated references were eliminated with this digital utility.

### 2.2. Eligibility Criteria

An ad hoc review team was assembled to carry out this systematic review. This team was composed of two specialists in oral medicine/pathology and molecular oncobiology (FFV and MPS). The studies that were selected passed through two stages. In the first phase, the two authors evaluated the titles and abstracts of the identified studies and discussed their selection in a consensus meeting. In the second phase, full-text studies were blindly evaluated by the same authors, and information was cross-referenced. Interobserver agreement was determined by Cohen’s kappa coefficient (κ) using the freeware Epidat 4.2. In cases of disagreement, the researchers recruited a third blind researcher to review the case (MEP).

Eligibility was set after inclusion criteria were met, such as: (i) original research studies published in English; (ii) evaluation of Caspase 3 or cleaved Caspase 3 expression using IHC in human tissues of patients affected by HNC or OPMD; (iii) analysis of the association between Caspase 3 overexpression and at least one of the following survival outcomes: overall survival (OS), disease-free survival (DFS) or disease-specific survival (DSS).

Exclusion criteria included: (i) non-English published studies; (ii) studies carried out on animal or in-vitro models; (iii) studies relating to pre-cancerous lesions not localized in the head or neck (e.g., skin lichen planus); (iv) studies about thyroid cancer; (v) pharmacological studies; (vi) comparisons between different groups or diseases; (vii) analyses of Caspase 3 gene polymorphisms; (viii) proteomic, transcriptomic or genomic-based research, different from IHC; (ix) studies with insufficient data to estimate hazard ratios (HR); (x) studies with duplicate cohorts. 

### 2.3. Data Extraction

The authors extracted data using a pilot tested form. This form included the following items: first author, year of publication, country where the study was conducted, sample size, staging edition used, recruitment period, tumor subsite, cut-off value for Caspase 3 IHC high–low expression, immunostaining pattern (nuclear/cytoplasmic), HRs for long-term outcomes with standard errors (SEs) or 95% confidence intervals (95% CIs).

### 2.4. Quality Assessment

An assessment of the risk of bias of the included studies was performed using parameters derived from the Reporting Recommendations for Tumor Marker Prognostic Studies (REMARK) [68] as previously reported by our group, the scale consisting of six parameters evaluating: (a) samples, (b) clinical data of the cohort, (c) IHC, (d) prognosis, (e) statistics and (f) classical prognostic factors. On the basis of the REMARK guidelines, each factor was considered as: adequate (A), inadequate (I) or not available (NA). Each item scored as A added one point to the overall quality assessment for each study. A score sheet was prepared for each included study, and quality scoring was independently undertaken by the authors (FFV and MPS). In the event of disagreement, the scores were discussed until a consensus was reached. Studies were categorized as high quality when the overall score was >4.

### 2.5. Statistical Analysis

The differences in Caspase 3 and cleaved Caspase 3 stainings were categorized as high, moderate, low or absent according to the cut-off chosen by original authors of the studies. HRs and 95% CIs were used as the measure of association in order to estimate the impact of Caspase 3 expression on the aforementioned long-term outcomes (OS, DFS and DSS). Multivariate or univariate HR values were used, but, when available, the former were chosen. When data on the HRs could not be directly traced, they were approximated using the methods described by Tierney et al. [69]. In cases where survival was available but a hazard ratio was not reported and not possible to estimate, an email was sent to the corresponding author. If the author did not respond, the data were excluded.

The pooled analyses were performed using the software Review Manager version 5.2.8 (Cochrane Collaboration, Copenhagen, Denmark; 2014). For survival analysis of Caspase 3, the natural logarithm of the HR and its SE were calculated and entered into the software. Q and I^2^ tests were employed to furtherly assess heterogeneity across studies. Presence of heterogeneity was considered significant for *p*-value < 0.05. According to assessment using the Higgins index, less than 30% was classified as low heterogeneity, between 30% and 60% as medium and over 60% as high. As a consequence, a fixed or random effect was calculated using the inverse of variance test, setting a *p*-value lower than 0.05 as the threshold of statistical significance. In addition, the results of the meta-analysis were summarized in forest plots. A funnel plot was further generated to visually inspect publication bias.

## 3. Results

### 3.1. Study Selection Process and Study Features

Through the search strategy, 91 records out of the 1536 records found were read full-text for eligibility. After full-text reading, 73 studies were excluded and 18 were included (Figure 2).

The value of the κ-statistic was 0.87, which indicates an excellent level of agreement between reviewers. A total of 1365 patients were analyzed in the 18 articles. Descriptive summaries of the included studies, listed chronologically from the oldest publication date, are displayed in Table 1. The data were collected from a period spanning 1970 to 2019, while the year of publication ranged from 2005 to 2019 [5,7,10,14,16,20,21,26,27,37,50,57,58,61,70,71,72,73]. Sample sizes ranged between 20 and 246 [5,7,10,14,16,20,21,26,27,37,50,57,58,61,70,71,72,73]. The studies were conducted in 14 different countries across Europe, South America and Asia. Caspase 3 and cleaved Caspase 3 expression were assessed in the cytoplasmic membrane in all the studies, and in one article cytoplasmic membrane and nuclear staining was observed [58]. The cut-off points for Caspase 3 and cleaved Caspase 3 expression varied among the studies, although 25% was the most frequently used [10,57,58,61,71]. The anti-Caspase 3 and cleaved Caspase 3 antibodies used were Caspase 3 (monoclonal) [7,14,26,27,57,58,61,70,72,74], Caspase 3 (polyclonal) [5,57,72], active Caspase 3 (monoclonal) [16,50,58,71] and active Caspase 3 (polyclonal) [10,21,37,57].

### 3.2. Quality Assessment within Studies

Regarding the analysis of Caspase 3 and cleaved Caspase 3 expression, a problem was commonly observed in relation to immunohistochemistry, due to the lack of specificity related to the antibodies used. Expression analyses in many cases were dubious, citing cleaved Caspase 3 when in relation to Caspase 3, or vice versa. Of the 18 studies, 8 performed a prognostic analysis. Eight studies fully complied with the REMARKS guidelines. According to aforementioned >4 cut-off point, all of these eight studies were considered of good quality (Appendix A [5,14,27,37,57,70,71,73]). Low risk of bias regarding clinical data was found in one study. Regarding the specific statistical analysis of each study, there was a notable risk of bias related to inadequate statistical analysis methods, erroneously reported data or lack of adjustments for confounding factors and intervals when related to the respective means.

### 3.3. Quantitative Evaluation (Meta-Analysis)

#### 3.3.1. Comparative Evaluation between Caspase 3 and Cleaved Caspase 3 Expression in OPMD and HNC

Results related to the comparison of Caspase 3 expression in OPMD and HNC showed a similar expression in neoplastic lesions, with an average of 51.9% (9.5–98.1) showing high/moderate expression in HNC, compared with 45.7% (14.6–84.7) in OPMD. Of interest is the fact that cleaved Caspase 3 showed a mean increase in HNC when compared with OPMD: 73.3% (38.6–88.3) versus 22.9% (7.1–38.7) of high/moderate staining, respectively. The standard deviation can be justified by the difference in cut-off, antibodies used, as well as the type, location, phenotype and other individual characteristics of each study (Figure 3; Appendix A [5,7,14,20,26,27,57,58,61,70,72,73]; Appendix A [10,16,21,37,50,57,71]).

#### 3.3.2. Quantitative Evaluation (Meta-Analysis)

A fixed effect model was used to evaluate the pooled HR, with a 95% CI for the outcomes of OS, DFS, and DSS on the basis of the presence or absence of substantial heterogeneity as shown by the *p*-values from their respective Q tests. The fixed effect pooled HR value (95% CI) of OS related to Caspase 3 IHC expression in the tissue of HNC patients was 1.48 (95% CI 0.95–2.28). Additionally, the rate of heterogeneity was low, as revealed by I^2^ = 31% (Figure 4). The fixed effect pooled HR value for DFS was 1.07 (95% CI 0.79–1.45). The rate of heterogeneity among the studies was negligible, as revealed by I^2^ = 0% (Figure 5). In the case of DSS, a fixed effect meta-analysis showed an HR of 0.88 (95% CI 0.69–1.12). Additionally, the heterogeneity was low, as revealed by I^2^ = 37% (Figure 6).

## 4. Discussion

It is currently known that proteins related to the apoptotic machinery have been identified as potential biomarkers, either for predicting prognosis or for developing drugs that can act on these genes [75]. This is because Hanahan and Weinberg in 2000 and 2011 described ten characteristics that provide a logical framework which allows us to understand how various stages of tumor pathogenesis occur in humans. One of the most important among these is the cancer cells’ ability to evade apoptosis [12,13]. 

Apoptosis is regulated by a balance of a multitude of proteins that play a role in inhibiting (e.g., Bcl-2, Bcl-x, mutant p53, survivin) or promoting (e.g., Bax, caspases) cell death [20,26,35,76]. Caspases are considered central regulatory proteins of apoptosis, and their expression servs as a marker of this process in several types of cancer [15,16,27,70,74]. However, results from this meta-analysis failed to highlight Caspase 3 expression as prognostic biomarker in patients with head and neck cancer.

With the increasing knowledge of molecular dynamics in cancer, many attempts have been made to identify more diagnostic and prognostic factors that may provide a clearer prediction of tumor behavior [37]. For example, the imbalance in the regulation of apoptotic molecules is responsible for the development of chemoresistance, and some of these can affect the clinical prognosis [5]. On the other hand, investigations of apoptotic cell numbers have shown inconsistency in HNC [77,78]. Different studies have found higher numbers of apoptotic cells in higher grade tumors, with higher cell proliferation and worse survival [79,80,81,82]. Apoptotic proteins can release growth stimulating signals to allow non-apoptotic tumor cells to proliferate and survive under stress conditions [61]. Indeed, in this inconclusive scenario, overexpression of Bcl-2, an anti-apoptotic protein, is reported to be associated with a better survival [83,84]. Huang et al. demonstrated that dying tumor cells use Caspase 3 as a mediator for growth-signals to enhance the repopulation of tumors undergoing radiotherapy. Their study also showed that patients with a high expression of Caspase 3 reported worse survival outcomes [57]. 

In support of previous statements, Singh et al. found a correlation of Caspase 3 expression with clinicopathological parameters, such as nodal involvement and staging [74]. On the other hand, Coutinho-Camillo et al. associated Caspase 3 expression with lower events of lymph node metastasis and advanced T-scores [16]. Similarly, Oudejans et al. observed a strong relation between absence of Caspase 3 activation and older age, higher T and N stage and failure to achieve complete remission [37]. Dozic et al. did not observe a statistical relationship between the expression of Caspase 3 parameters and clinical pathological variables such as sex, patient age, tumor location and tumor histology [27]. Andresakis et al. and Tanimoto et al. also reported no significant association [5,70]. Current evidence is contradictory and inconclusive, leaving the association of Caspase 3 with different clinic-pathological features in question.

The role of Caspase 3 in prognosis has also been debated. Silva et al. concluded that five-year overall and cancer-specific survival rates differed between patients with negative and positive Caspase 3 expression, agreeing with the observations of Tanimoto et al., who concluded that the five-year disease-specific survival rates of patients with and without Caspase 3 expression were 40.2% and 53.6%, respectively [14,70]. Similarly, in a study by Singh et al., high Caspase 3 expression showed a 30% drop in survival, while in Dozic et al. this difference was not evident [27,74]. 

Liu et al. concluded by univariate and multivariate analyses that Caspase 3 levels in tumor tissues were not associated with DSS or DFS in patients, and Andresakis et al. concluded that the prognostic significance of Caspase 3 expression was non-significant [5,71]. Of interest is the fact that when investigating cleaved Caspase 3, Liu et al. found that its expression was associated with poorer DFS in the univariate analysis [71]. In contrast, Oudejans et al. reported that the prognosis of patients with higher percentages of cleaved Caspase 3 positive tumor cells was favorable [37].

However, for Huang et al., expressions of cleaved Caspase 3 and Caspase 3 were not associated with DSS and DFS, but when patients were stratified by postoperative RT, high Caspase 3 expression was associated with poor DFS [57].

Although comparative studies are needed, this systematic review and meta-analysis is the first to investigate the expression of Caspase 3 and cleaved Caspase 3 in HNC and OPMD, as well as its association with the prognosis of these patients. Results from our systematic review and meta-analysis show that the expression of cleaved Caspase 3 is higher than that of Caspase 3 in HNC, while the opposite occurs in OPMD, prompting questions concerning the possibility of a failure in the cleavage of Caspase 3 in OPMD that could favor the malignancy process. We also observed that both Caspase 3 and cleaved Caspase 3 are higher in HNC than in OPMD, which seems contradictory because even with high levels of protein expression in HNC the apoptotic process does not occur sufficiently to control the progression of the disease. This goes against some authors who suggest that other mechanisms of evasion of apoptosis appear with the advancement of tumorigenesis.

We observed poor evidence of Caspase 3 expression in the prognosis of patients with HNC. Meta-analysis showed that the difference of expression of Caspase 3 is not involved in OS, DFS or DSS in patients with HNC, with an overall OS of 1.48 (95% CI 0.95–2.28, *p*-value = 0.08), DFS of 1.07 (95% CI 0.79–1.45, *p*-value = 0.66) and DSS of 0.88 (95% CI 0.69–1.12, *p*-value = 0.30).

In addition to being a pioneering review, our study’s strengths include the accumulation and use of studies with high quality scores and the inclusion of recently published articles to improve statistical power and prioritize studies that use precise analytical methods for reliable analysis of the expression. However, although the results of this systematic review and meta-analysis are supported by solid evidence, some limitations were observed when considering the studies individually, and these should be considered.

The main limitation observed was the absence of absolute values of many HRs [5,14,27,37,57,70], even in the presence of Kaplan-Meier with survival analysis. In these studies, data were extracted from the graphs using a protocol described in the methodology [69].

The heterogeneity of the lesions related to the different subsites where each study was focused, since HNC is a classification of tumors that includes a large group of anatomical structures, could be considered a limiting factor and may have generated divergences between the results for both expression analysis and survival analysis. For example, Dozic et al. showed 78% high/moderate Caspase 3 expression in a study focused on salivary gland carcinoma, significantly higher than that observed by Andressakis et al., which was 9.5% in tongue squamous cell carcinoma lesions [5,27].

The difference between the antibodies used could admittedly have generated deviations between the results, and in the case of the studies included in this meta-analysis, this was confirmed. Many authors showed concerning inconsistencies regarding the concepts of Caspase 3 and cleaved Caspase 3, claiming to have performed Caspase 3 analysis when instead antibodies referring to cleaved Caspase 3 had been used, e.g., the studies of Bascones-Ilundan et al., Coutinho-Camillo et al. and Bascones-Martínez et al. This could have generated a distortion in the results, since cleaved Caspase 3 is an active form of Caspase 3 [10,16,50]. In the same way, other authors used both antibodies for Caspase 3 and cleaved Caspase 3 where they made a comparison between both expressions, such as Huang et al., Hague et al. and Liu et al. In these cases, weighted averages were obtained between the results in order to maintain the reliability of the results of our study [57,58,71].

As well as the analysis of the expression of other proteins, it is known that they may vary among patients based on ethno–geographical distribution, which would explain some inconsistencies observed in the studies carried out in different parts of the world, and in our systematic review that includes studies performed in Europe [5,7,10,27,37,50,58,72], Asia [20,26,57,70,71,74] and South America [14,16,21,61], each of which use the same markers for cancer cell phenotypes and their association with clinical outcome.

Moreover, differences between the marking cut-off can also generate important deviations, since some authors considered the cut-off 25% [10,57,58,61,71], some 10% [50,74] and some 3% [70], while several authors did not specify the cut-off value [5,7,14,16,20,21,27,37,72].

Other limitations may be related to the differences between the number of samples, the counting method, the individual clinical characteristics of the tumors and the testing period, in addition to ambiguities in the distinction between OS, DFS and DSS, which may have influenced the differences in values observed between the articles. However, even with the limitations of this study, we believe in the reliability of the results, which will be widely applicable, although more immunohistochemical reports are still needed to validate this biomarker, under strict consensus and reliable methods, based on standardization of employed antibodies, scoring and target population.

## 5. Conclusions

Based on the results obtained in this study, we concluded that an increase in Caspase 3 seems to favor the progression of malignancy from OPMD to HNC. It is possible that a failure in the cleavage of Caspase 3 in OPMD could favor the malignancy process. The expression of Caspase 3 did not influence the prognostic values of survival in patients with HNC.

## Figures and Tables

**Figure 1 ijms-23-11937-f001:**
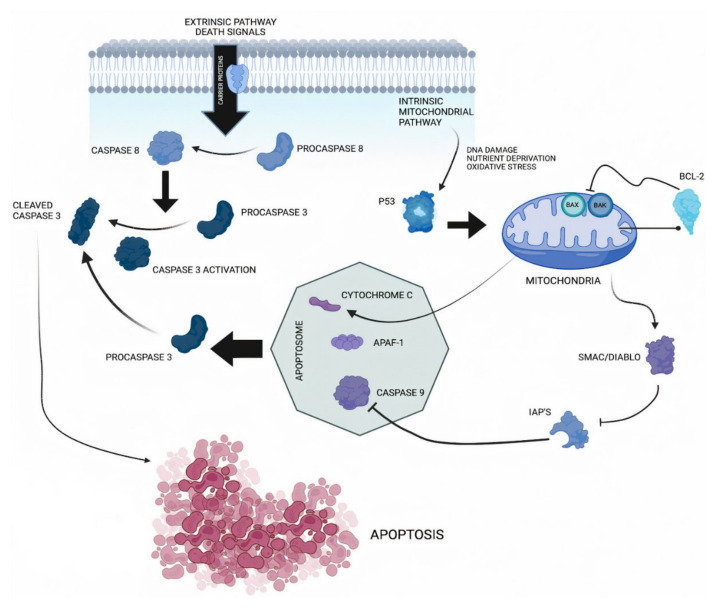
Caspase 3 pathway. The apoptotic process can be triggered by external or internal signals that lead to a common outcome. The extrinsic pathway can be activated when death ligands bind to their corresponding death receptor. Upon ligand binding, an adapter protein is recruited. This leads to the activation of Caspase 8. The intrinsic pathway is initiated by pathological intracellular processes such as DNA damage, nutrient deprivation or oxidative stress. Increased levels of pro-apoptotic proteins such as Bax or Bak (from the family of Bcl-2 proteins) may also be associated with this activation. This triggers the release of cytochrome c from mitochondria, which then binds to and activates apoptosis protease activator protein 1 (Apaf-1), which in turn binds to and activates Caspase 9. Active Caspase 8 or Caspase 9 triggers the cleavage of Caspase 3 through its proenzyme. For apoptosis to take place, inhibitory apoptosis proteins (IAP) must be inactivated by Smac/Diablo proteins.

**Figure 2 ijms-23-11937-f002:**
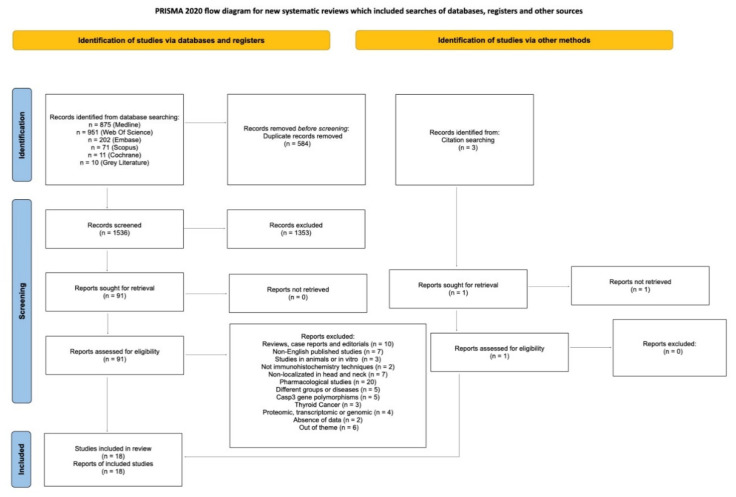
Flow diagram of literature search. Diagram according to Preferred Reporting Items for Systematic Reviews and Meta-Analyses (PRISMA) 2020.

**Figure 3 ijms-23-11937-f003:**
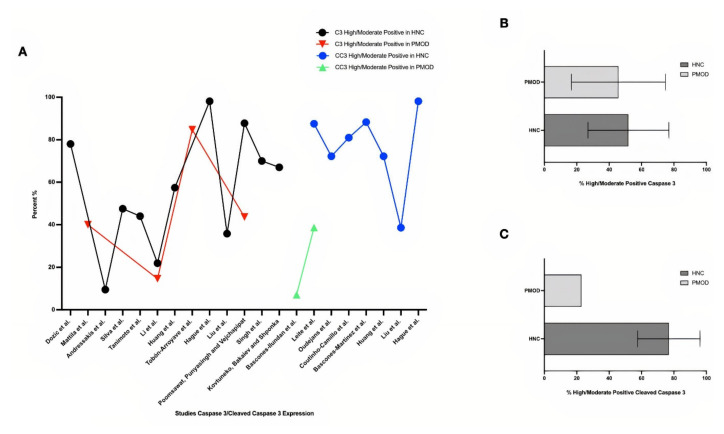
Graphical analysis of Caspase 3 and cleaved Caspase 3 expression in HNC and OPMD. (**A**) In an overview, the studies showed a higher percentage of Caspase 3 and cleaved Caspase 3 in HNC when compared with OPMD; however, the expression of cleaved Caspase 3 was more discrepant. (**B**) Caspase 3 expression showed average values of 51.9% in HNC and 45.7% in OPMD. (**C**) Cleaved Caspase 3 was 73.3% in HNC and 22.9% in OPMD [5,7,10,14,16,20,21,26,27,37,50,57,58,61,70,71,73].

**Figure 4 ijms-23-11937-f004:**
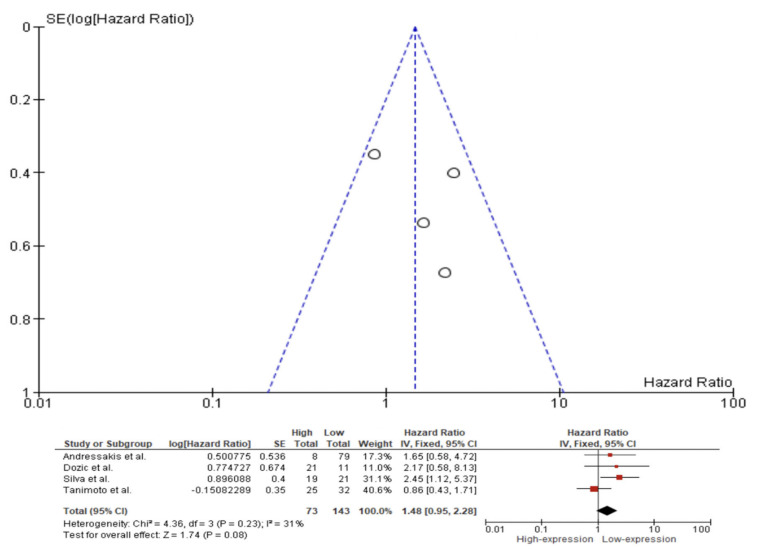
Forest plot and funnel plot for the association of higher Caspase 3 expression with overall survival. Squares represent study-specific hazard ratios; horizontal lines represent 95% confidence intervals (CIs); diamonds represent the overall hazard ratio estimate with its 95% CI [5,14,27,70].

**Figure 5 ijms-23-11937-f005:**
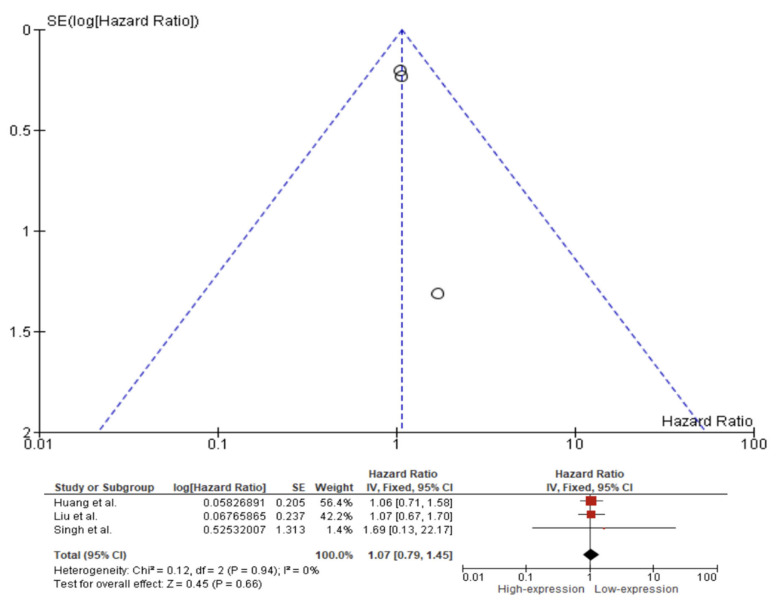
Forest plot and funnel plot for the association of higher Caspase 3 expression with disease-free survival. Squares represent study-specific hazard ratios; horizontal lines represent 95% confidence intervals (CIs); diamonds represent the overall hazard ratio estimate with its 95% CI [57,71,73].

**Figure 6 ijms-23-11937-f006:**
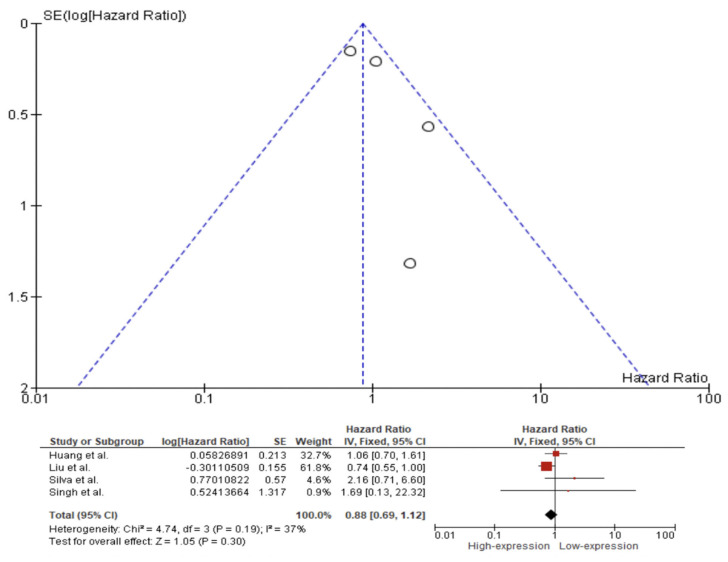
Forest plot and funnel plot for the association of higher Caspase 3 expression with disease-specific survival. Squares represent study-specific hazard ratios; horizontal lines represent 95% confidence intervals (CIs); diamonds represent the overall hazard ratio estimate with its 95% CI [14,57,71,73].

**Table 1 ijms-23-11937-t001:** Descriptive summaries of the included studies. NI = Non-Informed; UICC = Union for International Cancer Control; AJCC = American Joint Committee on Cancer; HNC = Head and Neck Cancer; OPMD = Oral Premalignant Disorders [5,7,10,14,16,20,21,26,27,37,50,57,58,61,70,71,72,73].

Reference Number	Author	Year	Country	Sample Size	Staging Edition	Tumor Subsite	Recruitment Period	Type of Lesion	Survival Analysis	Caspase 3/Cleaved Caspase 3 (Monoclonal)	IHC Pattern	Cut-Off Point (%)
-	Dozic et al. [27]	2016	Servia	50	NI	Salivary Glands	1998–2008	HNC	OS	CPP3 (monoclonal)	cytoplasmic membrane	NI
10.1016/j.archoralbio.2005.02.005.	Bascones-Ilundan et al. [50]	2006	Spain	52	NI	Oral Mucosa, Gingiva, Lip, Tongue	1999–2003	OPMD	UN	Active-Caspase 3 (monoclonal)	cytoplasmic membrane	10
10.1590/1678-775720160156.	Leite et al. [21]	2016	Brazil	120	NI	Oral Cavity, Lip	NI	HNC/OPMD	UN	Cleaved Caspase 3 (polyclonal)	cytoplasmic membrane	NI
10.1038/modpathol.3800398.	Oudejans et al. [37]	2005	Netherlands	36	UICC	Nasopharynges	1995–1996	HNC	OS	Active Caspase 3 (polyclonal)	cytoplasmic membrane	NI
10.1016/j.tripleo.2010.05.070.	Mattila et al. [72]	2010	Finland	66	NI	Oral Cavity	1991–2002	OPMD	UN	3CSP03 (polyclonal)	cytoplasmic membrane	NI
10.1017/S0022215108002636.	Andressakis et al. [5]	2008	Greece	87	NI	Tongue	1998–2006	HNC	OS	3CSP03 (polyclonal)	cytoplasmic membrane	NI
10.1002/hed.21602	Coutinho-Camillo et al. [16]	2010	Brazil	229	NI	Oral Cavity	1970–1992	HNC	UN	Cleaved Caspase 3 (monoclonal)	cytoplasmic membrane	NI
10.1002/hed.25763	Silva et al. [14]	2019	Brazil	40	NI	Salivary Glands	2014–2019	HNC	DSS, OS	Caspase 3 (monoclonal)	cytoplasmic membrane	NI
10.4317/medoral.18901.	Bascones-Martínez et al. [10]	2013	Spain	41	UICC	Oral Cavity	NI	HNC	UN	Cleaved Caspase 3 (polyclonal)	cytoplasmic membrane	25
10.1016/j.canlet.2004.11.049	Tanimoto et al. [70]	2005	Japan	57	UICC	Oral cavity, Oropharynx, Hipopharynx	1989–2000	HNC	OS	Caspase 3 (monoclonal)	cytoplasmic membrane	3
10.3892/ol.2017.6626	Li et al. [20]	2017	China	45	NI	Oral Cavity	2005–2007	HNC/OPMD	UN	Caspase 3 (monoclonal)	cytoplasmic membrane	NI
10.18632/oncotarget.20494.	Huang et al. [57]	2017	Taiwan	185	AJCC	Buccal Mucosa	1993–2006	HNC	OS, DFS, DSS	Cleaved Caspase 3 and Caspase 3 (polyclonal)	cytoplasmic membrane	25
10.1046/j.1601-0825.2003.00998.x.	Tobón-Arroyave et al. [61]	2004	Colombia	30	NI	Oral Cavity	NI	OPMD	UN	CPP32 (monoclonal)	cytoplasmic membrane	25
10.1371/journal.pone.0180620.	Liu et al. [71]	2017	Taiwan	246	AJCC	Tongue	1991–2010	HNC	DFS, DSS	Caspase 3 and Cleaved-Caspase 3 (monoclonal)	cytoplasmic membrane	25
10.1002/path.1630	Hague et al. [58]	2004	UK	54	NI	Tongue, Labial Mucosa, Buccal Mucosa, Palate, Floor of Mouth, Alveolar Process/Gingiva	NI	HNC/OPMD	UN	Caspase 3 and Cleaved-Caspase 3 (monoclonal)	cytoplasmic and nuclear membrane	25
10.1097/PAI.0b013e31828a0d0c.	Poomsawat, Punyasingh and Vejchapipat [26]	2014	Thailand	104	NI	Oral Cavity	2000–2009	HNC/OPMD	UN	Caspase 3 AF835 (monoclonal)	cytoplasmic membrane	NI
10.3233/CBM-190149.	Singh et al. [73]	2019	India	20	AJCC	Oral Cavity	NI	HNC	DFS, DSS	Caspase 3 31A1067 (monoclonal)	cytoplasmic membrane	10
10.36740/WLek201912108	Kovtuneko, Bakaiev and Shponka [7]	2019	Ukraine	80	AJCC	Maxillary Sinus	2011–2016	HNC	UN	Caspase 3 (monoclonal)	cytoplasmic membrane	NI

## Data Availability

All data generated or analyzed during this study are included in this published article (and its Appendix A files).

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
