# Peer review of "Caspase 3 and Cleaved Caspase 3 Expression in Tumorogenesis and Its Correlations with Prognosis in Head and Neck Cancer: A Systematic Review and Meta-Analysis"

_ijms, 2022, doi:10.3390/ijms231911937_

Round 1
Reviewer 1 Report
This is a nice review and meta-analysis article, concerning the correlation of Caspase 3 and Cleaved Caspase 3 Expression in oral Tumorigenesis and prognosis of head and neck cancer. Therefore, I suggest it for publication. Two remarks for the authors; (1) Please look for more references in the literature, if available and (2) a minor remark is highlighted in the attached text.

Author Response
Manuscript ID: ijms-1920110
Title: Caspase 3 and Cleaved Caspase 3 Expression in Tumorogenesis and Its Correlations with Prognosis in Head and Neck Cancer: A Systematic Review and Meta-Analysis.
International Journal of Molecular Sciences
Dear Editor,
We would like to submit the revised manuscript entitled “Caspase 3 and Cleaved Caspase 3 Expression in Tumorogenesis and Its Correlations with Prognosis in Head and Neck Cancer: A Systematic Review and Meta-Analysis.” to the Editorial Board of the International Journal of Molecular Sciences. We would like to thank the reviewers for their in-depth review of the manuscript. We agree with the comments which have been made and we have revised our manuscript accordingly. After having implemented the proposed edits we are confident of the overall presentation and clarity of our manuscript. Thank you very much for your attention and we look forward to hearing from you at your earliest convenience.
Yours sincerely,
Dr. Fábio França Vieira e Silva
Reviewers' comments:
Reviewer 1 comments:
- This is a nice review and meta-analysis article, concerning the correlation of Caspase 3 and Cleaved Caspase 3 Expression in oral Tumorigenesis and prognosis of head and neck cancer. Therefore, I suggest it for publication. Two remarks for the authors; (1) Please look for more references in the literature, if available and (2) a minor remark is highlighted in the attached text.
- First, thank you for your comments about our article and for considering a good meta-analysis regarding the correlation of Caspase 3 and Cleaved Caspase 3 Expression in oral Tumorigenesis and prognosis of head and neck cancer. We really appreciate it a lot.
- Regarding references, we agreed that there was a limited number of references and that we could expand them to improve the bibliography on the topic indicated by our systematic review. 35 new references were added according to the markings in the manuscript.
- We agree with the remark indicated in the text and include the word "usually" in the sentence: "Head and neck cancer (HNC) is an ascending disease and includes tumors 'usually' arising from the epithelia of different anatomical origins...".
Reviewer 2 comments:
- (1) The introduction and discussion should be focused more on the observations and novelty of this study. More concluding remarks must be also added; (2) Provide high-quality Figures and tables.
- Regarding the first comment, we agree that this greater focus on demonstrating the relevant novelties and aims of the study should be prioritized. Therefore, we decided to include already in the introduction, two new paragraphs indicating what our study has of relevance and novelty, focusing on the possibilities of findings from the proposed meta-analysis.
- We apologize for the quality of the previously uploaded images, which were at 300dpi. All the figures have been improved and we've upped the image quality to 600dpi to make sure they're right for the journal. We also changed the shapes of the tables to make sure of the texts indicated are more understandable.

Reviewer 2 Report
1. The introduction and discussion should be focused more on the observations and novelty of this study. More concluding remarks must be also added.
2. Provide high-quality Figures and tables
Author Response

(The authors gave the same response as above.)

Round 2
Reviewer 2 Report
Accept